# Enhanced astrocyte responses are driven by a genetic risk allele associated with multiple sclerosis

Gerald Ponath[1], Matthew R. Lincoln[1,2,3], Maya Levine-Ritterman[1], Calvin Park[1], Somiah Dahlawi[1], Mayyan Mubarak[1], Tomokazu Sumida [1,2,3], Laura Airas[4], Shun Zhang [5], Cigdem Isitan[1], Thanh D. Nguyen[5], Cedric S. Raine[6], David A. Hafler[1,2,3] & David Pitt[1]

Epigenetic annotation studies of genetic risk variants for multiple sclerosis (MS) implicate dysfunctional lymphocytes in MS susceptibility; however, the role of central nervous system (CNS) cells remains unclear. We investigated the effect of the risk variant, rs7665090[G], located near *NFKB1*, on astrocytes. We demonstrated that chromatin is accessible at the risk locus, a prerequisite for its impact on astroglial function. The risk variant was associated with increased NF-κB signaling and target gene expression, driving lymphocyte recruitment, in cultured human astrocytes and astrocytes within MS lesions, and with increased lesional lymphocytic infiltrates and lesion sizes. Thus, our study establishes a link between genetic risk for MS (rs7665090[G]) and dysfunctional astrocyte responses associated with increased CNS access for peripheral immune cells. MS may therefore result from variant-driven dysregulation of the peripheral immune system and of the CNS, where perturbed CNS cell function aids in establishing local autoimmune inflammation.

[1] Department of Neurology, Yale School of Medicine, New Haven, CT 06511, USA. [2] Department of Immunobiology, Yale School of Medicine, New Haven, CT 06511, USA. [3] Broad Institute of MIT and Harvard University, Cambridge, MA 02141, USA. [4] Division of Clinical Neurosciences, University of Turku, Turku 20520, Finland. [5] Department of Radiology, Weill Cornell Medical College, New York, NY 10021, USA. [6] Department of Pathology, Albert Einstein College of Medicine, Bronx, NY 10461, USA. Correspondence and requests for materials should be addressed to D.P. (email: david.pitt@yale.edu)

Multiple sclerosis (MS) is a genetically mediated inflammatory disease of the central nervous system (CNS) in which infiltrating immune cells lead to focal, demyelinating lesions[1]. Genome-wide association studies (GWAS) have now identified over 200 genetic variants that confer increased risk of developing MS[2]. A recent epigenetic annotation study has demonstrated that MS risk variants are highly enriched in immune enhancers active in T and B cells[3], suggesting that risk variant-mediated MS susceptibility is driven by changes in gene regulation in lymphocytes. This was confirmed in our recent work where we prioritized 551 potentially associated MS susceptibility genes, and found that they implicated multiple innate and adaptive pathways distributed across different cells of the immune system[2]. It remains unclear whether genetic variants only affect immune cells or whether they also impact on CNS cell function, thus driving MS risk by dysregulating CNS-intrinsic pathways.

We addressed this question by investigating how a common MS risk variant, rs7665090[G], which is relevant to NF-κB signaling, changes astrocyte function. This risk variant has an approximate frequency of 55% in the general population, and increases the odds ratio for MS susceptibility by 1.09 per G allele carried[4], and has recently been shown to substantially increase NF-κB p50 expression and NF-κB activation in lymphocytes[5].

As a master regulator of innate and adaptive immunity, NF-κB plays a critical role in autoimmunity, including in MS, where 18% of all allelic MS risk variants are estimated to affect or intersect with the NF-κB signaling pathway[4]. NF-κB signaling also plays a role in the activation of astrocytes, a cell type which is critically involved in the formation of MS white matter lesions. Astrocytes are uniquely positioned to recruit myelin-specific lymphocytes into the CNS, as they regulate blood brain barrier (BBB) permeability and, in an activated state, become a principal source of chemoattractants[6,7]. In addition, astrocytes control lesion-promoting functions of other constituent cell types of MS lesions, and can themselves become neurotoxic by reducing homeostatic functions and adopting a phenotype toxic to oligodendrocytes and neurons[8]. Astrocyte-specific inhibition of NF-κB activation has been shown to dramatically ameliorate immune infiltration and tissue damage in experimental autoimmune encephalomyelitis (EAE), an animal model of MS, suggesting that genetic modulation of NF-κB signaling may alter astrocyte responses and astrocyte-mediated MS lesion pathology[9,10].

Here, we set out to determine the impact of the MS risk variant rs7665090[G] on astrocyte function both in cell culture and in reactive astrocytes within MS lesions. Our results provide compelling evidence that genetic MS risk is linked to the dysregulation of astrocyte function.

## Results

**Chromatin accessibility in the *NFKB1* risk haplotype block.** We assessed chromatin accessibility within the *NFKB1-MANBA* risk haplotype block tagged by rs7665090 with an assay for transposase-accessible chromatin with high throughput sequencing (ATAC-seq) on magnetic bead sorted unstimulated and stimulated human fetal and iPSC-derived astrocytes. For comparison, we used ex vivo, unstimulated human effector and regulatory T cells[11]. This haplotype block spans the intergenic region downstream of the *NFKB1* gene and a part of the *MANBA* gene. Open chromatin sites captured by ATAC-seq were present in the intergenic region in both astrocytes and T cells (Fig. 1), suggesting that the risk haplotype block-associated increase in *NFKB1* gene expression that has been previously shown in T cells[5] may also apply to astrocytes. In contrast, chromatin at a risk locus within the intragenic region of *NFKB1*, which is associated with non-CNS autoimmune diseases (inflammatory bowel disease and systemic scleroderma), was accessible only in T cells, but not in astrocytes (Supplementary Fig. 1). The specificity of our dataset

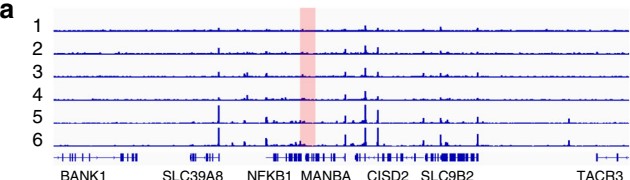

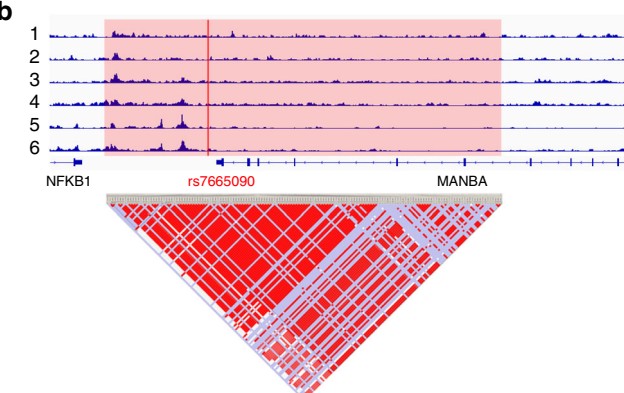

**Fig. 1** Chromatin accessibility in the *NFKB1/MANBA* risk haplotype block in astrocytes and T cells. The rs7665090 risk variant tags a haplotype block that lies within *MANBA* and the intergenic space between *NFKB1* and *MANBA* and contains more than 90 variants in tight linkage disequilibrium[32,33]. **a** Normalized ATAC-seq profiles in the *NFKB1/MANBA* risk locus are similar in unstimulated and stimulated human iPSC-derived astrocytes (1, 2), unstimulated and stimulated human fetal astrocytes (3, 4), and in ex vivo effector and regulatory T cells (5, 6). The height of the bar graph at each point represents the number of unique, singly-mapping reads at each genomic position; each track is normalized for library size. The *NFKB1/MANBA* haplotype block is shaded in red. The genomic coordinates are chr4:101,796,383–103,694,281. **b** Higher magnification (chr4:102,614,123–102,674,492) of the risk haplotype block, shaded in red, shows accessible chromatin, mostly in the intergenic region between *NFKB1* and *MANBA* with comparable ATAC-seq profiles in astrocytes and T cells. The location of the tagging SNP rs7665090 is indicated by the red line. Pairwise linkage disequilibrium (*D'*) values between SNPs in the CEU population of the 1000 Genomes Project are indicated below (red indicates *D' > 0.95*)

was further confirmed by the mutually exclusive chromatin accessibility in lineage-specific genes of astrocytes and T cells (glial fibrillary acidic protein (GFAP), CD2). Thus, our results indicate that while *NFKB1* is expressed by a multitude of cell types, accessibility of different regulatory enhancer regions for the *NFKB1* gene varies among different cell types.

**Effect of the rs7665090[G] variant on human astrocytes.** Next, we determined the impact of the rs7665090[G] risk variant on NF-κB signaling in iPSC-derived astrocytes generated from fibroblasts of 12 MS patients and healthy controls, that are homozygous either for the risk (rs7665090[GG]) or protective variant (rs7665090[AA]) (Supplementary Fig. 2). We controlled for potential effects of the following risk variants that are predicted to intersect with the canonical NF-κB signaling pathway: rs1800693 (*TNFRSF1A*), rs12296430 (*LTBR*), rs4810485 (*CD40*), rs1077667 (*TNFSF14*), and rs12148050 (*TRAF3*). The majority of iPSC lines were heterozygous for the additional NF-κB relevant risk variants (Supplementary Fig. 3), suggesting that these variants did not interfere with the rs7665090[G] effect on astrocyte phenotype in our experiments.

In unstimulated astrocytes, NF-κB signaling was low in both groups, as measured by the degradation of inhibitor of NF-κBα

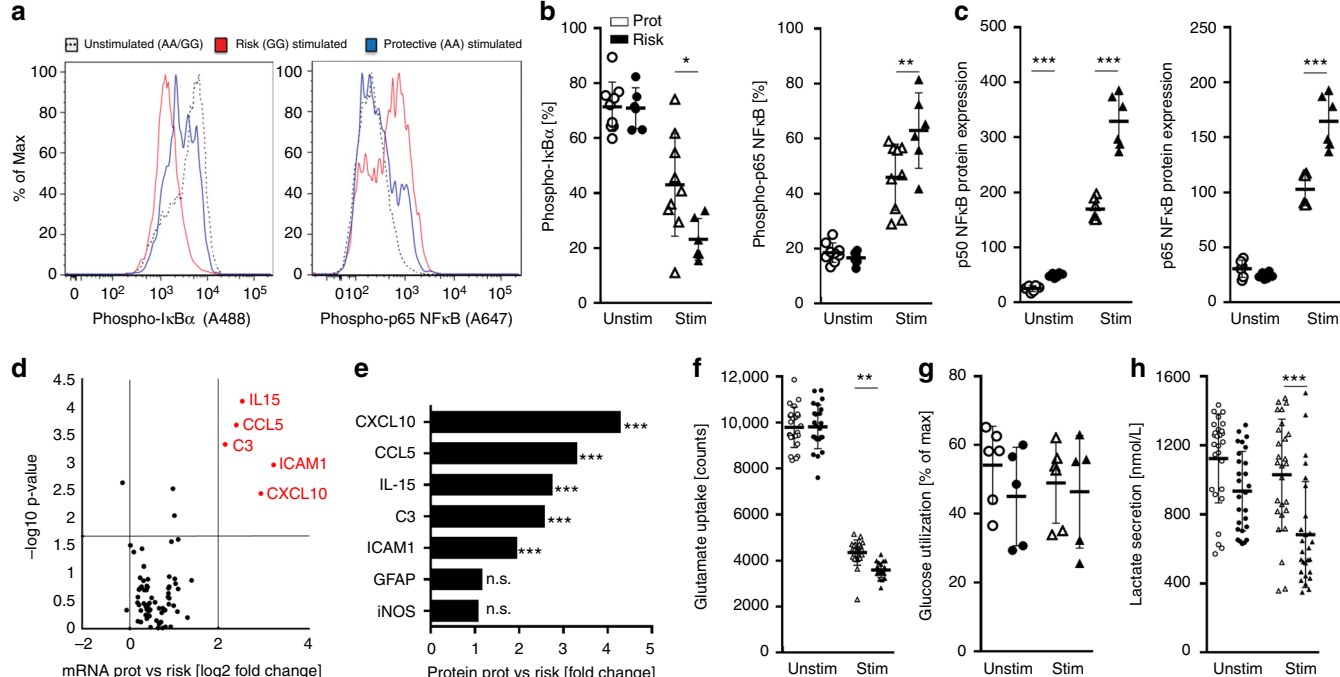

**Fig. 2** rs7665090[G] variant effect on human iPSC-derived astrocytes. **a, b** Degradation of IκBα and phosphorylation of p65 in iPSC-derived astrocytes with the risk or protective variants at a resting state and 10 min after stimulation with TNFα (50 ng/ml) and IL-1β (10 ng/ml) by flow cytometry. **c** Expression of p50 and p65 in unstimulated and stimulated (50 ng/ml TNFα and 10 ng/ml IL-1β for 48 h) iPSC-derived astrocytes from both groups by western blot. **d** Volcano plot profiling astroglial expression of 84 NF-κB target genes after stimulation with TNFα, IL-1β, and IFNγ for 16 h. Red dots indicate genes with ≥2-fold expression and FDR-adjusted *p*-values ≤ 0.05. **e** Protein expression corresponding to genes with significantly increased expression. **f** Uptake of L-[3,4-³H] glutamic acid by iPSC-derived unstimulated and stimulated (with TNFα and IL-1β) astrocytes with the risk or protective genotype. **g, h** Glucose uptake and lactate secretion by astrocytes with the risk or protective variants. Data represents means ± s.d. from three independent experiments. *p*-Values shown for the post-hoc Tukey–Kramer test performed after one-way ANOVA. *$p < 0.05$, **$p < 0.01$, and ***$p < 0.001$

(IκBα) and phosphorylation of p65. Treatment of astrocytes with a combination of TNFα and IL-1β, cytokines that induce or enhance NF-κB signaling and play a major role during MS lesion development[12,13], increased NF-κB activation in both groups; however, NF-κB activation was significantly higher in astrocytes carrying the risk variant (Fig. 2a, b). In addition, at a resting state, expression of NF-κB p50, but not of p65, was higher in astrocytes with the risk compared to the protective variants. After stimulation, p50 and p65 expression was substantially upregulated in both groups, but significantly higher in astrocytes with the risk variant (Fig. 2c).

We then examined the effects of the risk variant on the expression of a panel of 84 NF-κB target genes. TNFα, IL-1β, and IFNγ stimulation resulted in significant upregulation of 23 genes in astrocytes with the protective variant, and 28 genes in astrocytes with the risk variant compared to baseline (Supplementary Fig. 4A). Transcripts that were differentially expressed by 2-fold or higher ($p \leq 0.05$) in stimulated astrocytes with the risk compared to the protective variants were IL-15, ICAM1, CXCL10, CCL5, and complement component 3 (C3), after correction for multiple comparisons (Fig. 2d; Supplementary Fig. 4B-D). The upregulation of protein expression was confirmed with ELISA or western blot (Fig. 2e). These findings indicate that the rs7665090 risk variant activates a specific set of NF-κB target genes in TNFα/IL-1β/IFNγ-stimulated astrocytes that facilitates lymphocyte recruitment and activation. Furthermore, risk variant-associated upregulation of C3 suggests astrocytic polarization towards a recently described toxic phenotype, termed A1 in analogy to M1 macrophages, for which C3 is the main marker[8].

The rs7665090 variant has been reported to have an eQTL effect on *MANBA* expression in transformed fibroblasts and skin

tissue[14] and *MANBA* has been assigned as a gene candidate to the rs7665090[G] effect in MS[4]. We found that *MANBA* expression in iPSC-derived astrocytes was comparable in both groups and unchanged after pro-inflammatory stimulation, suggesting that the rs7665090 risk variant impact on *MANBA* expression is cell type-specific and/or stimulus-dependent (Supplementary Fig. 4E).

We subsequently investigated whether the risk variant impacts on homeostatic and metabolic functions of astrocytes, namely glutamate transport and uptake/release of glucose and lactate. Glutamate uptake was robust in unstimulated iPSC-derived astrocytes in both groups, and deteriorated with pro-inflammatory stimulation, as described previously[15]. The removal of glutamate was marginally but significantly more impaired in iPSC-derived astrocytes with the risk compared to the protective genotype (Fig. 2f). Glucose uptake did not differ between stimulated and unstimulated astrocytes with either variant (Fig. 2g), while the release of lactate was slightly but significantly diminished in astrocytes with the risk variant after stimulation, but not at baseline (Fig. 2h).

**rs7665090[G] variant effect on astrocytes in MS lesions**. Having determined the effect of the rs7665090 risk variant on astrocytes in vitro, we wanted to know whether comparable phenotypic changes can be seen in reactive astrocytes in MS lesions. From a total of 14 MS autopsy cases from our brain bank, homozygous for the risk or the protective variant, we identified 10 cases that contained chronic active white matter lesions, in which we examined the variant impact on astrocytic phenotypes. As with the iPSC lines, we genotyped the MS cases for additional risk variants that could interfere with NF-κB signaling (Supplementary Fig. 3, 5).

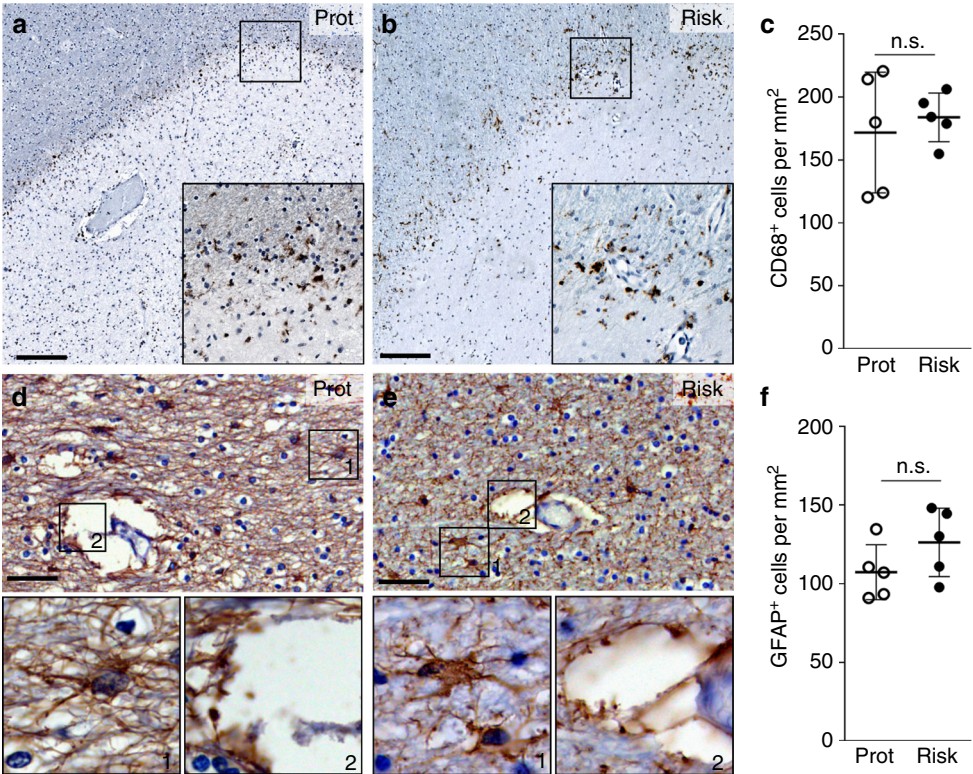

**Fig. 3** Glial density in MS lesions with the rs7665090 risk and protective genotypes. **a, b** Staining for CD68 (brown) shows activated microglia at the lesion rim of chronic active lesions, counterstained with hematoxylin (blue). Magnifications in the inset. **c** Quantification of CD68+ cell densities in the rim of 29 lesions (15 risk group, 14 protective group) from 10 MS cases (five cases per group) shows no significant difference between groups. The number of lesions from each case ranged from 1 to 5. **d, e** Staining for GFAP (brown) shows hypertrophic astrocyte morphology (inset 1) and partially disrupted astrocytic glia limitans (inset 2) at the lesion rim. **f** Quantification of GFAP+ cell densities in 10 lesions from 10 MS cases (five cases per group) shows no significant difference between groups. Dots represent average cell densities for all lesions per case. Data represent means ± s.d. Significance was determined with unpaired *t*-test. Scale bars = 100 μm

To obtain lesions with similar inflammatory activity for our subsequent studies, we selected active lesions with comparable CD68+ cell densities at the lesion rim (Fig. 3a–c). Using bright-field imaging of sections stained with GFAP, we found that astrocytes in the active lesion rim were uniformly hypertrophic, with comparable densities in both groups (Fig. 3f). Similarly, morphological assessment of the glia limitans did not suggest differences in astroglial end-feet coverage of blood vessels between the risk and protective groups (Fig. 3d, e). In contrast, confocal imaging of immunofluorescent-labeled sections demonstrated high, uniform expression of the NF-κB p50 and p65 subunits in the cytosol and nucleus of hypertrophic astrocytes, which was significantly elevated in astrocytes with the risk variant (Fig. 4a, b). Hypertrophic astrocytes in lesions with the risk variant showed enhanced immunofluorescent staining for CXCL10, CCL5, IL-15, ICAM1, and C3d (Fig. 4c, d; Supplementary Fig. 6A), the same target genes that were upregulated in iPSC-derived astrocytes. Moreover, the expression of these markers was clearly separated with little overlap between both groups (Supplementary Fig. 6B). NF-κB target genes (CCL2 and CXCL1) and NF-κB independent markers of astroglial activation (GFAP, iNOS) that were not differentially regulated in iPSC-derived astrocytes also did not differ between lesions with the risk or protective variants (Fig. 4d and Supplementary Fig. 6A, B).

Both p50 and p65 were near absent in non-reactive astrocytes in the lesion vicinity and normal appearing white matter (NAWM), indicating low constitutive levels of NF-κB signaling in non-activated astrocytes (Supplementary Fig. 6C). We did not stain for phosphorylated p65, as phosphorylation decays rapidly in post-mortem tissue, and is therefore not a reliable indicator of NF-κB activation[16]. Enhanced immunoreactivity for NF-κB p50 and p65 in lesions with the risk variant was not restricted to astrocytes, but was also found in activated microglia in the lesion rim in the vicinity of blood vessels (Fig. 4e, f). In contrast, expression levels of NF-κB p50 and p65 in endothelial cells did not differ between both groups. Moreover, p50/65 was present only in the cytosol, suggesting absence of NF-κB activation (Fig. 4g, h and Supplementary Fig. 6D, E).

**rs7665090G variant effect on lesion pathology and lesion load.** Since increased astroglial expression of CXCL10, CCL5, and ICAM1 implies enhanced recruitment of lymphocytes, we quantified number of infiltrating, perivascular lymphocytes within MS lesions, and lesion sizes, in autopsy cases from both groups in a total of 29 lesions. We found that the number of perivascular CD3+ T cells per blood vessel in the active rim was significantly higher in lesions with the risk than the protective variants, but not in the gliotic lesion center (Fig. 5a–d; Supplementary Fig. 7). In addition, the ratio of perivascular CD3+ T cells to CD68+ microglia at the rim was higher in lesions in the risk group, as was overall lesion size (Fig. 5e, f).

Correlation studies showed strong positive correlations between NF-κB signaling in astrocytes and expression of chemokines, CCL5 and CXCL10 ($r = 0.88$; $r = 0.84$), while no correlation was found between NF-κB signaling and GFAP/iNOS expression in astrocytes or CD68+ cell density at the lesion rim

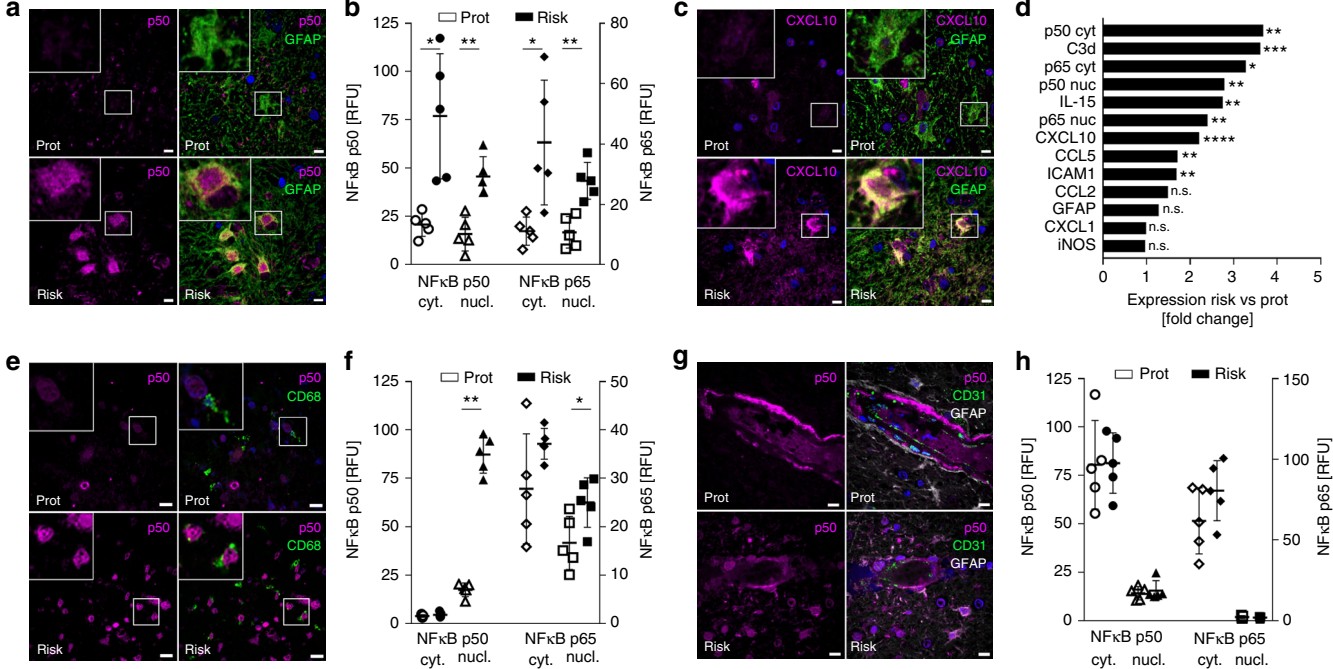

**Fig. 4** Effect of the rs7665090[G] variant on NF-κB signaling in white matter MS lesions. **a** Confocal microscopy images of hypertrophic astrocytes at the lesion edge, labeled with fluorescent antibodies against p50 (magenta) and GFAP (green) and counterstained with Hoechst 33342. **b** Densitometric quantification of NF-κB p50 and p65 in the cytosol/nucleus of hypertrophic lesional astrocytes (5 cases per group). **c** Hypertrophic astrocytes stained with fluorescent antibodies against CXCL10 (magenta) and GFAP (green) and counterstained with Hoechst 33342. **d** Differential expression of immune mediators and activation markers in astrocytes from both groups. **e** Activated microglia at the lesion edge labeled with fluorescent antibodies against p50 (magenta) and CD68 (green) and counterstained with Hoechst 33342. **f** Densitometric quantification of NF-κB p50 and p65 in the cytosol/nucleus of activated microglia (5 cases per group). **g** Endothelial cells at the lesion edge labeled with fluorescent antibodies against p50 (magenta) and CD31 (green) and counterstained with Hoechst 33342. **h** Densitometric quantification of NF-κB p50 and p65 in the cytosol/nucleus of endothelial cells (5 cases per group). Data represent means ± s.d. p-Values shown for the post-hoc Tukey–Kramer test after one-way ANOVA, *p < 0.05 and **p < 0.01. Scale bars = 15 μm

(Fig. 5g, h, Supplementary Fig. 8). Similarly, we found strong correlations between CD3$^+$ cell infiltration and astroglial expression of cytosolic p50 ($r = 0.80$) and CXCL10 ($r = 0.67$), and between lesion sizes and astroglial expression of C3d ($r = 0.81$) and IL-15 ($r = 0.88$) (Fig. 5g, i, j, Supplementary Fig. 8).

Finally, we investigated whether the histological finding of increased lesion sizes in cases with rs7665090 risk variant, can be confirmed in MS patients by quantifying lesions with MRI. Since individual lesions often become confluent in advanced MS and are thus difficult to delineate, we quantified the overall lesion load per patient rather than individual lesion sizes. We recruited a total of 134 patients homozygous for the risk or protective variant from the MS clinics at Yale, USA and Turku, Finland (78 and 56 patients, respectively, Supplementary Fig. 9). White matter lesion load on fluid-attenuated inversion recovery (FLAIR) images was determined using an automated segmentation algorithm[17], followed by manual correction by three independent raters. We found that the risk variant was not associated with a significant increase in lesion load in patients (Supplementary Fig. 10).

## Discussion

In the present study, we examined the impact of a risk variant for MS susceptibility, rs7665090[G], on astrocyte function both in vitro and in active MS lesions, as well as on lesion pathology. We demonstrated that the risk variant is associated with enhanced astroglial NF-κB signaling and upregulation of a subset of NF-κB target genes that drive lymphocyte recruitment and neurotoxicity, as implied by increased expression of C3[8]. We observed an expression shift only for a relatively narrow spectrum of NF-κB

target genes, presumably because we examined only one cell type (astrocytes) under specific stimulatory conditions. Homeostatic and metabolic functions of astrocytes were moderately affected by the risk variant, but may contribute to cellular damage in lesions by inducing excitotoxicity and metabolic uncoupling from axons/ neurons. Thus, with our results, we are linking genetic susceptibility for MS to changes in astrocyte function. Because astrocytes play a major role in controlling leukocyte traffic into the CNS[18], excessive upregulation of chemokines and adhesion molecules in astrocytes is likely to increase accessibility for peripheral immune cells to the CNS, which may lower the threshold for MS lesion formation. The increased CNS accessibility is demonstrated by the enhanced lymphocytic infiltration in MS lesions with the risk genotype. On histological sections, the risk variant was also associated with increased lesion sizes, which was highly correlated with astrocyte expression of C3, a measure of astroglial neurotoxicity, and of IL-15, an activator of cytotoxic CD8 T cells. This effect was less pronounced than the variant effect on NF-κB signaling, because multiple additional factors are likely to influence lesion size. Similarly, the total white matter load in MS patients is probably driven by a multitude of genetic and environmental factors, which could explain why the lesion load was not affected by the rs7665090 risk variant.

One limitation of our study is that we did not directly examine the risk variant effect on lesion formation in a murine model. This is because the genetic architecture of non-coding regulatory regions of *NFKB1* differs considerably between humans and mice, and because the causative variant in the rs7665090-tagged haplotype block is unknown. The complex genome engineering that is required to replace large genomic fragments in mice by their human counterparts is beyond the scope of this study.

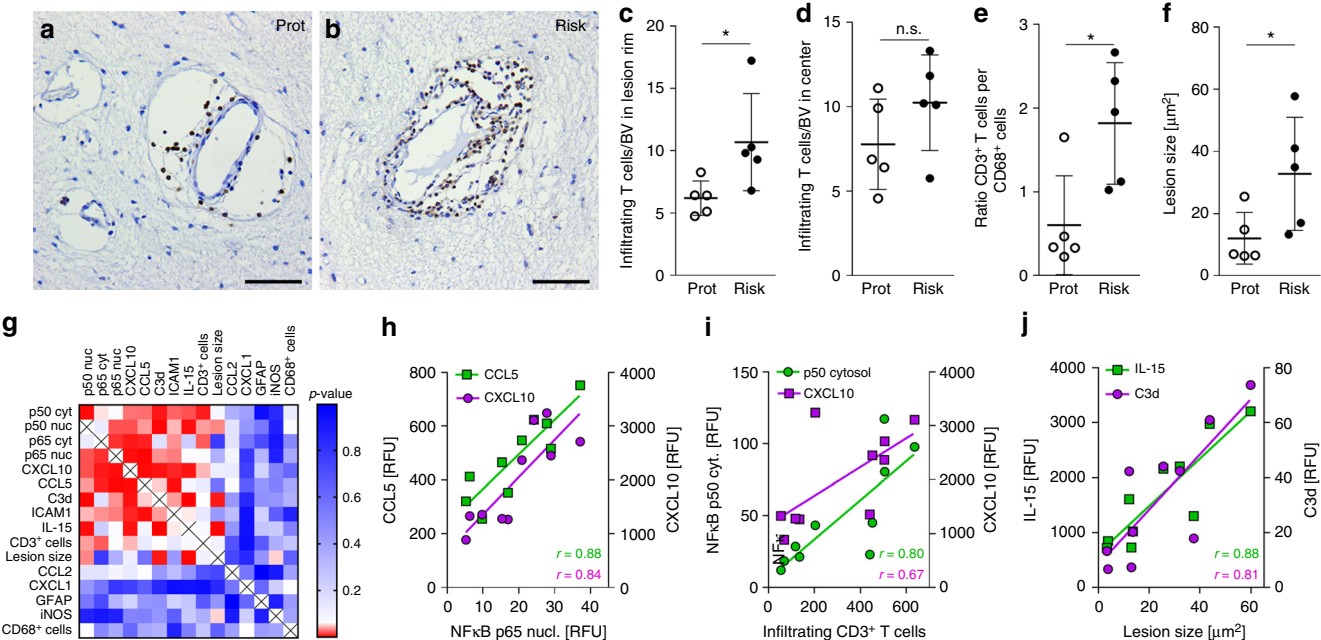

**Fig. 5** Effect of the rs7665090$^G$ variant on MS lesion pathology. **a**, **b** Bright-field images of perivascular infiltrates within chronic active MS lesions labeled with anti-CD3 antibody. Scale bar = 50 μm. **c**, **d** Quantification of infiltrating, perivascular CD3$^+$ T cells per blood vessel in the rim (**c**) and center (**d**) of 29 lesions (15 risk group, 14 protective group) from 10 MS cases (5 cases per group) shows a significant difference between groups in the rim ($p = 0.0413$), but not in the center ($p = 0.1943$). Dots represent average CD3$^+$ T cells per blood vessel for all lesions of a single case. The number of lesions from each case ranged from 1 to 5. **e** Ratios of perivascular CD3$^+$ T cells to CD68$^+$ microglia in the rim show a significant difference between groups ($p = 0.0199$). **f** Quantification of lesion sizes, determined as area of demyelination, shows a significant difference between groups ($p = 0.0491$). Dots represent average size for all lesions of a single case. Data represents mean ± s.d. $p$-Values shown for the unpaired $t$-test. $*p < 0.05$. **g** Heatmap of computed Pearson correlation coefficients from expression values of 5 cases per group (red, $p$-value < 0.05; blue, $p$-value > 0.05). **h** Correlation of NF-κB p65 nuclear expression with CCL5 ($r = 0.88$, $p = 0.001$) and CXCL10 ($r = 0.84$, $p = 0.003$) expression in astrocytes. **i** Correlation of infiltrating CD3$^+$ T cell with cytosolic NF-κB p50 ($r = 0.80$, $p = 0.005$) and CXCL10 ($r = 0.67$, $p = 0.036$) expression. **j** Correlation of lesion size with IL-15 ($r = 0.88$, $p = 0.001$) and C3d ($r = 0.81$, $p = 0.005$)

We also demonstrated that the impact of the rs7665090$^G$ variant on NF-κB signaling and NF-κB target gene expression in astrocytes is substantial, consistent with other studies that report large variant effects on individual cellular pathways[5,19,20]. This robust increase in NF-κB activation contrasts with the minor increases in MS susceptibility conferred by rs7665090 and other risk variants outside of the major histocompatibility complex[4]. This indicates that individual pathways, even if strongly dysregulated, play only a minor part in overall MS susceptibility. Since individual patients carry a unique or near-unique combination of risk variants and environmental exposures, discrete sets of biological pathways are dysregulated in each patient, which causes the considerable heterogeneity in clinical presentation and treatment responses in MS. This may explain why the rs7665090 risk variant substantially impacts on NF-κB target gene expression but not on disease outcomes such as lesion load.

Furthermore, the variant-associated changes in lesion pathology are not mediated exclusively by astrocytes. We demonstrated that NF-κB signaling was also upregulated in microglia/macrophages in lesions with the rs7665090 risk variant, as was previously shown for lymphocytes[5]. Thus, enhanced lymphocytic infiltration and lesion sizes in MS lesions are likely caused by a combination of enhanced responses of astrocytes, microglia, and lymphocytes, which mutually reinforce the effects in individual cell types. The variant does, however, not have an effect across all cell types involved in lesion formation, as NF-κB expression in endothelial cells was equal in both groups.

Thus, our results suggest that MS risk is not only driven by dysregulated immune cells, but also by dysfunctional cellular responses from astrocytes and microglia, which help establish autoimmune inflammation in the CNS. An additional argument for a role of astrocytes in genetically mediated MS risk is that the rs7665090$^G$ variant also confers increased risk for primary biliary cholangitis (PBC), an autoimmune disease of the liver associated with lymphocytic infiltration[21]. The rs7665090$^G$ variant mediated risk for PBC may be driven by excessive NF-κB signaling in hepatic stellate cells, an important cell type in the pathophysiology of PBC, that exhibit striking morphological and functional similarities to astrocytes, including expression of GFAP, regulation of the blood–tissue barrier, and NF-κB-dependent recruitment of phagocytic Kupffer cells[22,23]. Conversely, NF-κB related variants located in chromatin regions that are not accessible in astrocytes (rs1598859 and 3774959), are associated with risk for inflammatory bowel disease and systemic scleroderma but not MS, suggesting that these variants do not drive astrocyte-mediated lymphocyte recruitment to the CNS.

In summary, our study provides evidence for the first time that genetic variants associated with MS risk directly perturb CNS cell functions. It remains to be shown that risk variants have an impact on other CNS-constituent cell types, and possibly on non-immune pathways, which may plausibly lead to increased MS susceptibility. To this end, a systematic correlation of MS risk variants with detailed chromatin landscapes of CNS cells may help delineate the potential relevance of risk variants in different CNS cell types and help identify CNS-intrinsic disease-causing pathways. Two recent studies on human microglia found that candidate genes associated with MS risk alleles were preferentially expressed in microglia[2,24], further strengthening the idea that CNS-intrinsic mechanisms contribute to MS susceptibility. The presence of genetic variants may have implications for therapeutic

approaches in MS patients. Patients that carry one or several risk variants within a given pathway may benefit from therapies that specifically target this pathway.

## Methods

**Study design.** The aim of this study was to establish the effect of the rs7665090 risk variant on (i) astrocyte function in vitro and in situ, (ii) on MS lesion pathology, and (iii) on lesion load in MS patients. iPSC-derived astrocytes, autopsied MS cases, and MS patients were all homozygous for either the risk or protective variant. We used astrocyte cultures derived from a total of 24 iPSC lines obtained from 6 MS patients and 6 healthy donors. For histological lesion analysis, we examined a total of 39 MS lesions from 14 MS patients. 29 lesions were used to determine lesion sizes and the degree of CD3 infiltration; 10 chronic active lesions were selected for their comparable CD68$^+$ cell density and used to determine astrocyte responses. Finally, we identified 134 MS patients homozygous for the risk or protective variant at the MS Clinics in Yale, USA and Turku, Finland, to determine their lesion load on MRI. For patients who were treated with disease-modifying therapies such as natalizumab, rituximab, ocrelizumab, and alemtuzumab[18–20], which are highly effective in preventing lesion formation, we defined disease duration from onset of symptoms until the initiation of these treatments, thereby accounting for treatment history. Primary data of patients' characteristics are provided in Supplementary Fig. 9. In a separate study, we identified 127 MS patients homozygous for the risk or protective variant at the MS Clinics in Yale and Turku to determine their total brain volume on MRI. Sample sizes were dictated by the availability of MS autopsy cases and of genotyped MS patients. In astrocyte culture, we performed at least three independent experimental replicates. From MS autopsy tissue, we selected lesions with comparable CD68$^+$ densities at the lesion rim and subsequently determined expression levels from at least 20 reactive astrocytes, activated microglia, and endothelial cells per lesion. Analysis was performed in all experiments in a blinded fashion.

The study complied with all relevant ethical regulations. Human CNS tissue, skin biopsies, and MRI images were obtained according to Institutional Review Board-approved protocols (Yale Human Investigation Committee) and informed consent was obtained from all human participants.

**Generation of iPSCs and differentiation into astrocytes.** We obtained skin punch biopsies from six MS patients at the Yale MS Clinic that were homozygous for the rs7665090 risk variant or protective variant (Supplementary Fig. 5) and generated explant cultures for the derivation of primary human fibroblasts as described[25]. Fibroblasts were cultured by Tempo Bioscience, Inc. (San Francisco, CA, USA; http://www.tempobioscience.com), reprogrammed to three individual iPSC colonies per patient and expanded, using their proprietary reprogramming protocol. Pluripotency of all 18 iPSC lines was confirmed by assessing the expression of biomarkers including Oct4, Tra-1-80, Nanog, and SSEA4. iPSC colonies were differentiated into astrocyte progenitors and matured into astrocytes, expressing S100beta and GFAP, using Tempo Bioscience's serum-, feeder-, integration-, and genetic elements-free, non-viral technology. For maturation, astrocytes were plated on poly-L-ornithine/laminin and differentiated with DMEM/F12/neurobasal medium (50%/50%) containing 10 ng/ml BMP-4 for 2–4 weeks.

Six iPSC lines were obtained from the New York Stem Cell Foundation and differentiated into neural stem cells using the STEMDiff$^{TM}$ SMADi Neural Induction Kit (Stem Cell Technologies). Neural stem cells were further differentiated into astrocytes using Astrocyte Medium (ScienCell Research Laboratories) for 40 days. In experiments using iPSCs from both sources, astrocytes from Tempo Biosciences were also differentiated with Astrocyte Medium (ScienCell Research Laboratories) to ensure similar conditions. After differentiation, >95% of the cells were GFAP$^+$ and GLAST$^+$ as determined by flow cytometry (Supplementary Fig. 12). Magnetic bead-sorted GLAST$^+$ astrocytes were used for all experiments.

**Human fetal astrocyte culture.** Human fetal astrocytes were obtained from Thermo Fisher Scientific (Gibco® Human Astrocytes) and cultured in basal medium (DMEM), N-2 supplement, and 10% fetal bovine serum according to the manufacturer's recommendation.

**Chromatin accessibility profiling.** Human fetal astrocytes ($n = 2$) and iPSC-derived astrocytes ($n = 4$) were profiled for chromatin accessibility using the assay for transposase-accessible chromatin (ATAC-seq)[11,26]. Aliquots of 5000 cells were incubated with transposase solution containing 1% digitonin[26] at 37 °C with agitation at 300 rpm for 30 min. After transposition, DNA was purified (MinElute PCR Purification Kit; QIAGEN) and transposed fragments were minimally PCR amplified[11] and purified using the Agencourt AMPure XP system (Beckman Coulter). The average fragment size was estimated by Bioanalyzer (Agilent) and libraries were quantitated with the qPCR-based Library Quantification Kit (KAPA Biosystems). Purified libraries were sequenced on the Illumina HiSeq 2000, generating paired-end 100 bp fragments. Fragments were aligned to hg38 with bowtie2[27] and unique, singly mapping reads were retained for further analysis.

Visualization tracks were generated by calculating the number of reads aligning at each genomic position and normalizing for library size.

**Gene expression profiling in iPSC-derived astrocytes.** Astrocytes were stimulated with 50 ng/ml TNFα, 10 ng/ml IL-1β, and 100 U/ml IFNγ for 16 h before total RNA extraction (RNeasy Micro Plus Kit; Qiagen), reverse transcription (RT$^2$ First Strand Kit, Qiagen) and gene profiling with a Human NF-κB Signaling Targets RT$^2$ Profiler PCR Array (#PAHS-025Z, Qiagen), according to the manufacturer's instructions[28]. The gene expression of GFAP, NOS2, and MANBA was performed using TaqMan gene expression assays (#Hs00909233_m1, #Hs01075529_m1, #Hs01099178_m1; Thermo Fisher) according to the manufacturer's instructions.

**Flow cytometry, western blot, and ELISA.** To assess NF-κB activation with flow cytometry, astrocytes were stimulated with 50 ng/ml TNFα and 10 ng/ml IL-1β for 10 min, dissociated with accutase, stained with fluorescent-labeled primary antibodies (Supplementary Fig. 11), and analyzed on either a FACSCalibur or a LSRII flow cytometer (BD Biosciences)[28]. Protein expression in unstimulated and stimulated (50 ng/ml TNFα and 10 ng/ml IL-1β, 48 h) cultured astrocytes were determined with western blot as previously described[28] with primary antibodies listed in Fig. S6. Proteins were visualized with an enhanced chemiluminescence using an ImageQuant LAS 4000 camera (GE Healthcare). Densitometry was performed with ImageJ software and values normalized to loading control (β-actin) were used for analyses. Cytokine release was quantified with sandwich ELISAs (DuoSet ELISA for CXCL10, CCL5, IL-6, R&D Systems; complement C3 ELISA; Abcam). Supernatants from unstimulated and stimulated astrocyte cultures were collected, centrifuged, and assayed according to the manufacturer's instructions.

**Metabolic assays.** Glucose uptake was measured in unstimulated and stimulated astrocytes (50 ng/ml TNFα and 10 ng/ml IL-1β; 48 h) via incorporation of the fluorescent glucose analog 2-NBDG (Thermo Fisher Scientific). 5 μM 2-NBDG was added to cells in low glucose medium (1 g/L D-Glucose) for 30 min. After washing with HBSS, intracellular fluorescence was determined with an Infinite M1000 fluorescent plate reader (Tecan). Lactate secretion was determined from stimulated and unstimulated cell culture supernatants using an enzymatic assay, which produces a colorimetric (570 nm)/fluorometric ($\lambda_{ex} = 35$ nm/$\lambda_{em} = 587$ nm) product proportional to lactate content. (Sigma Aldrich). To quantify glutamate uptake, we incubated astrocyte cultures with HBSS buffer containing 0.5 μM L-glutamate and L-[$^3$H] glutamate (1 μCi; PerkinElmer) at a 100:1 ratio for 5 min at 37 °C. Cells were rapidly moved onto ice, washed twice with ice-cold glutamate-free HBSS buffer, and lysed with 0.1 N NaOH solution. [$^3$H] radioactivity was measured using a scintillation counter and counts were normalized to total protein levels per sample[29].

**Genotyping of formalin-fixed autoptic brain tissue.** Formalin-fixed CNS tissue of 82 MS patients were obtained from the PI's CNS bank and the Colorado Brain Bank. Since most cases from the PI's MS tissue bank came to autopsy in the 1980s/1990s, prior to the era of effective MS therapies, the majority of patients did not receive standard disease-modifying treatments.

CNS tissue was genotyped by isolating DNA using the DNeasy Blood and Tissue Kit and QIAamp DNA FFPE Tissue Kit (both Qiagen). Pre-amplification of DNA and genotyping for rs7665090 was performed with the TaqMan PreAmp Master Mix Kit and a TaqMan genotyping assay (Applied Biosystems). Genotyping was carried out in duplicates and repeated at least three times in separately obtained DNA samples. We identified 8 cases with the rs7665090 risk genotype and 6 cases with the protective genotype. In addition, CNS tissue and iPSC lines were tested for the following NF-κB relevant risk variants with the putative gene targets: rs1800693 (TNFRSF1A), rs12296430 (LTBR), rs4810485 (CD40), rs1077667 (TNFSF14), and rs12148050 (TRAF3) (details in Supplementary Fig. 3). None of these patients have received treatment with high-efficiency therapies, including natalizumab, rituximab, ocrelizumab, or alemtuzumab.

**Bright-field immunohistochemistry and MS lesion classification.** For lesion characterization, tissue blocks containing white matter lesions were sectioned, quenched with 0.03% hydrogen peroxide, incubated with primary antibodies against MBP (myelin), CD68 (myeloid cells), GFAP (astrocytes), and CD3 (lymphocytes), processed with the appropriate biotinylated secondary antibody and avidin/biotin staining kit with diaminobenzidine as chromogen (Vector ABC Elite Kit, DAB Kit, Vector Laboratories), and counterstained with hematoxylin[28]. Controls included isotype antibodies for each primary antibody. White matter lesions were categorized as acute, chronic active, and chronic silent. In 29 chronic active lesions, we quantified the density of CD68$^+$ cells at the lesion rim and CD3$^+$ lymphocytes within perivascular infiltrates throughout the lesion area. 10 representative chronic active lesions (five cases per group) with comparable CD68 densities were selected

**Immunofluorescent labeling of MS lesions**. To visualize protein expression in reactive astrocytes within lesions, sections were incubated with primary antibodies, listed in Supplementary Fig. 11, overnight at 4 °C, processed with HRP-conjugated secondary antibodies for 2 h at RT and reacted with Alexa Fluor tyramide (Life Technologies) for 10 min. Subsequently, sections were dyed with 0.7% Sudan Black and $CuSO_4$ to quench auto-fluorescence and counterstained with DAPI. Sections were examined and images acquired on an UltraVIEW VoX (PerkinElmer) spinning disc confocal Nikon Ti-E Eclipse microscope using the Volocity 6.3 software (Improvision). Images were processed with the ImageJ software[30]. Cytosolic and nuclear expression of NF-κB p50 and p65 were quantified by densitometric analysis of fluorescent immunoreactivity of GFAP[+]-positive hypertrophic astrocytes, CD68[+]-microglia, and CD31[+]-endothelial cells. Chemokine expression levels of CCL2, CCL5, CXCL1, CXCL10, as well as ICAM1, IL-15, complement factor C3, iNOS, and GFAP were quantified of GFAP-positive hypertrophic astrocytes at the lesion rim. Acquisition and analysis were performed in a blinded manner.

**MRI acquisition and T1/T2 FLAIR lesion segmentation**. MRI scans were acquired on a 3 T Siemens Skyra scanner (Yale) and 3 T Philips Ingenuity TF PET/MR scanner or a Philips Gyroscan Intera 1.5 T Nova Dual scanner (Turku). Total brain volumes, derived from non-Gadolinium enhanced T1 sequences, were calculated using mixed voxel segmentation and normalized to head size[31]. Automated lesion detection was performed on T2 weighted FLAIR sequences with the LPA algorithm available as part of the lesion segmentation toolbox (LST) (http://www.applied-statistics.de/lst.htm), followed by manual correction using itk-SNAP software version 3.x by three reviewers blinded to the patients' genotype.

**Statistical analysis**. Data represent means ± standard deviation from three independent experiments. Group comparisons of two samples were carried out by unpaired Student's $t$-tests. Comparisons of groups of ≥4 were analyzed by one-way ANOVA followed by the Tukey–Kramer multiple comparison test. In the figures, we have reported only the $p$-values for differences between the risk and protective groups. For multiple comparisons in the gene expression study of 84 NF-κB target genes, $p$-values, computed by unpaired Student's $t$-test, were adjusted for false discovery rate using the Benjamini–Hochberg procedure. For correlation analysis, Pearson correlation coefficients were computed. All values passed the D'Agostino-Pearson omnibus normality test for Gaussian distribution. *$p < 0.05$, **$p < 0.01$, ***$p < 0.001$, and ****$p < 0.0001$.

## Data availability

The datasets generated during and/or analyzed during the current study are available from the corresponding author on reasonable request.

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

## Acknowledgements

D.P. is supported by the National Multiple Sclerosis Society (RG-1610-26049), by the National Institutes of Health grant R01 NS102267, and by a generous gift from Stanley Trottman. D.A.H. is supported by National Institutes of Health grants P01 AI045757, U19 AI046130, U19 AI070352, and P01 AI039671, and the Nancy Taylor Foundation for Chronic Diseases. M.R.L. is supported by an endMS Postdoctoral Fellowship Award from the Multiple Sclerosis Society of Canada. T.S. is supported by the Banyu Fellowship Program and Uehara Research Fellowship Program. L.A. is supported by the Finnish Academy, the Sigrid Juselius Foundation, and a Grant for

Multiple Sclerosis Innovation by MerckSerono. T.D.N. is supported by the National Multiple Sclerosis Society grant RG-1602-07671 and the National Institutes of Health grant R01 NS090464. Sequencing service was conducted at Yale Stem Cell Center Genomics Core facility, which was supported by the Connecticut Regenerative Medicine Research Fund and the Li Ka Shing Foundation. The authors would like to acknowledge the Rocky Mountain Multiple Sclerosis Center Tissue Bank for contributing post-mortem MS brain tissue for this study. This investigation was supported in part by a grant from the National Multiple Sclerosis Society (RG-1610-26049).

## Author contributions

Conceptualization and study design: D.P. and G.P. Experiments and data analysis: G.P., M.R.L., M.L.-R., C.P., S.D., M.M., T.S., S.Z., C.I., and T.D.N. Supervision and resources: D.P., L.A., C.S.R., and D.A.H. Writing: G.P., M.R.L., and D.P.

## Additional information

**Competing interests:** The authors declare no competing interests.

