## [Peer Review File · Nature Communications]

Reviewers' Comments:

Reviewer #1:

Remarks to the Author:

The manuscript „Enhanced astrocyte responses are driven by a genetic risk allele associated with Multiple Sclerosis“ by Ponath et al explores the role of the MS risk allele rs7665090G for astrocytes, CNS resident cells with important functions in MS pathology. Using elaborate in vitro assays combined with post mortem histological analyses, the authors conclusively describe the mechanistic basis for enhanced astrocyte pathology induced by the rs7665090G risk variant, namely enhancement of NF-kB dependent pathways. The most important findings are that chromatin at the risk locus is accessible in fetal human astrocytes, a prerequisite for rs7665090G driven pathology in astrocytes. Second, the risk variant translates to functional effects mediated by enhanced expression and activity of NF-kB p50 and p65 as well as their target genes *IL15*, *Icam1*, *Cxcl10*, and *Ccl5*. Interestingly, in iPSC derived astrocytes, rs7665090G also causes metabolic changes such as decreased glutamate uptake and lactate secretion, pathways relevant for neurotoxicity. Fourth, comparing post mortem MS tissue of risk allele patients to controls, the authors show increased NF-kB p50/p65 expression and NF-kB target gene expression in astrocytes as well as enhanced lymphocyte recruitment in chronic active MS lesions. Finally, MRI studies reveal that the risk locus correlates with increased T2 lesion volumes in MS patients.

General critique:

The study presented by Ponath et al presents the relevance of a genetic risk allele for astrocyte biology in a concise and convincing manner; defining the alterations induced by the presence of the risk allele, it underlines the importance of NF-kB dependent pathways in CNS astrocytes for MS pathogenesis. This work is highly innovative and of high novelty and importance, as it is among the first to define the effects of genetic risk variants in CNS resident cells. Its findings are relevant for both basic scientists studying MS pathogenesis as well as for clinicians treating MS patients.

Thus, I have a few points to consider in a revised version only. If these points are addressed, I recommend publication of this important study.

1. In the results section, ATAC sequencing both from human fetal and iPSC derived astrocytes is mentioned. However, in Figure 1 only fetal astrocytes are shown. Since iPSC derived astrocytes are used for the subsequent studies and the comparability of iPSC derived astrocytes to fetal (or even adult) astrocytes is under scientific debate, showing the comparison of fetal and iPSC derived astrocytes in the ATAC Seq would increase the conclusiveness of the study.
2. How were human fetal astrocytes generated? Inclusion of this information in the Materials and Methods section will be appreciated.
3. Please show representative stainings for GFAP and GLAST in the iPSC cultures before and after sorting for GLAST positive cells to underline the effectiveness of the differentiation and selection procedure.
4. In Figure 3D, NF-kB p50 and p65 are globally increased in the risk variant astrocytes both in the cytosol and nucleus. Is there also an increase in the nuclear translocation of NF-kB in rs7665090G? Analyzing the ratio of nuclear to cytoplasmic NF-kB p50 and p65 would be interesting to determine potential increased activity of NF-kB in rs7665090G astrocytes.

Minor points:

1. In the results section, Fig S2 is referred to as Fig. S7 in the results section. Please adjust.
2. Table S5: please use generic names instead of trade names.

Reviewer #2:

Remarks to the Author:

Ponath et al use cellular assays and immunostaining of autopsy material to demonstrate that

human derived-astrocytes which express an MS risk variant, rs7665090-G located between the NFKB1 gene and MANBA gene, lead to upregulation of NF-kB expression, protein activity, and target gene expression, as well as markers of A1 reactive astrocytes described by the Barres group. By comparison of iPSC-derived astrocytes and immunostaining from patients homozygous for the risk allele v. protective MS risk variant allele, the authors show that astrocytes plays a critical role in MS pathogenesis through upregulation of a pathway previously implicated in immune cells primarily due to expression of a genetic risk factor allele. However, whether the role of these astrocytes is reactive or causative to lesions seen on MRI and pathology is not directly addressed in this study.

Major issues:

1. Astrocytes are not the only cell that comprises the blood brain barrier; did they also study the MS risk variant effect on pericytes or endothelial cells (at least on autopsy material?) If not, they should still include a short discussion of the potential contributors to the blood brain barrier. Are there any disruptions in astrocyte end-feet morphology in the specimens from patients with the MS Risk variant gene?
2. The one sentence summary says 'astrocyte responses that promote lesion formation' but they do not provide data demonstrating formation of any lesions.
3. Regarding the sentence "the risk haploblock-associated increase in NF-kB signaling, shown previously in T cells (ref 5), may also apply to astrocytes". The cited study looks at variants proximal to the NFKB gene; what is the potential mechanism for how a variant in the region distal to the NFKB1 can alter its expression?
4. Why do patients who are homozygous for the protective allele still develop MS? What is the difference in astrocyte role and function that the authors would predict based on this data?
5. Page 6, end of paragraph 1 – should cite 'data not shown' for non-reactive astrocytes in the lesion vicinity. In addition, it would be helpful to define the density of reactive v non-reactive astrocytes present in the lesions examined, and whether there was a difference between risk variant genotype. How does this compare to MS patients without either allele?
6. Figure S3 – GFAP staining should be shown separate from as well as merged with the second label (like p65). In Fig 3A, what is the p65+ cell identity adjacent to the GFAP+p65+ cell? How many p65+GFAP+ v. p65+GFAP-v. p65-GFAP+ cells (and similarly for all the other markers) are there in the lesions? Rather than densitometric analysis as shown in S3B, the proportion of positive v. negative cells relative to the population of astrocytes in the lesion would be a stronger argument for altered astrocyte signaling in MS risk variant-expressing astrocytes.
7. For the autopsy source patients, which ones were on Tysabri (treatment is not listed in Table S7) – could the protective variant-expressing patients have been on Tysabri, and therefore have fewer lymphocytes in the lesions?
8. Figure 4B: are these numbers of T cells normalized (such as to area of lesion?) If they are not normalized, these comparisons may be misleading.
9. The risk variant is only one of many that may be contributing to the phenotype observed. The authors should examine the other NFkB variants mentioned for their composite effect or calculate an NFkB risk score from them.
10. The C3 marker of A1 astrocytes was associated with neurotoxicity and not inflammation. The authors should examine T1 black holes and brain volumes in the same cohort.
11. The discussion should be expanded to include how altered astrocytes could 'promote' or possibly maintain or worsen lesions in MS. There is no data in this paper that supports astrocytes playing a role in formation of lesions that is worsened by expression of the risk variant. What could be the next steps to understand this role?
12. The markers for A1 astrocytes are increased in risk-variant expressing astrocytes; have the authors addressed whether microglia expressing the MS-risk variant are altered in promoting formation of A1 astrocytes (or whether they are also different within the MS lesion pathology) based on the work of Liddelov et al?

Reviewer #3:

Remarks to the Author:

In this manuscript, Ponath and colleagues investigate the role of the DNA polymorphism rs7665090G on astrocyte function. This variant has been consistently associated with risk to multiple sclerosis, and given the predominantly immune nature of other MS associations, authors wish to explore whether variation at this position alters CNS functions in any way. They start by showing that rs7665090G increases NF- κ B signaling in astrocytes and consequently drives lymphocyte recruitment in vitro and in pathological specimens. They also show that this risk allele is associated with increased lymphocyte infiltration within lesions in MS patients. Authors conclude that rs7665090G alters astrocyte function resulting in increased CNS access for peripheral immune cells.

This study is rigorously conducted and conclusions are logically derived from results. Still, some shortcomings were identified:

1. Fig1: The assessment that the chromatin accessibility profiles are "similar" is subjective. Can authors show statistics here (KS test or similar)? They also must show that the profiles for other (non-immune or CNS-related) cell types is substantially different.
2. Fig 2: no correction of multiple comparisons (n=84) is attempted. It looks like only IL15 would survive this. Thus, if this is the only evidence presented, conclusions on the involvement of C3, while potentially interesting, remains speculative.
3. Fig 2: Fig 2E repeats information from fig 2A. Another volcano plot at protein level would represent the same information and reduce the Figure size by one panel.
4. If 18% of MS associations affect NF- κ B, isn't it surprising to see such a large effect with just 1 variant?
5. Where else (outside of astrocytes) might this variant have an effect? Their observation that enhancer immunoreactivity for NF- κ B in lesions with the risk genotype was not restricted to astrocytes (also found in GAFFP- cells) supports the possibility that this variant affects NF- κ B function in a much broader set of cells.

Reviewer #4:

Remarks to the Author:

In their expression analysis the authors find that the level of p50 is higher in iPSC derived astrocytes from risk allele homozygotes (n=3) than in protective allele homozygotes (n=3), and also find modestly significant differential expression for a few of the 84 NFKB1 target genes they tested. While these are interesting and novel data, I am not sure that the authors suggestion that "the rs7665090 risk variant activates a SPECIFIC SET of NFKB target gene" can be justified. Given that the study subjects will inevitably have varied at MS risk genotypes other than rs7665090 it could be that effects on other NFKB1 target genes have been confounded. Similarly, the modest power of their analysis, which included just 6 subjects, is also likely to have confounded the chances of seeing effect on other NFKB1 target genes. Previous expression studies have highlighted a prominent eQTL effect of this variant on MANBA expression but have not highlighted effects on NFKB1. This could of course be an astrocyte specific effect on NFKB1 but it's a shame the authors did not measure the expression of MANBA in these astrocytes.

In their pathological assessments the authors observed equivalent changes in hypertrophic astrocytes from carefully selected lesions examined histologically in 10 biopsied cases. However, they do not seem to have tested other NFKB1 target genes from their list of 84; at least they only present data for those genes highlighted in their iPSC expression analysis. The authors also report nominally significant associations of the rs7665090 genotype with pathological and MRI features. However, these are only nominally significant and I am not exactly sure I follow the logic behind excluding patients with low lesion load and short duration of disease. I can't see why these should be biased with respect to genotype. The authors should include an analysis based on the scans from all 93 subjects at least to reassure the reader that the excluded group did not include an over

representation of the risk allele carriers just by chance. Even if the genotype associations with pathology and MRI features were confirmed this would not imply that these effects were driven by NFKB1 changes in astrocytes. The mechanism suggested by the author, that the risk allele increases the expression of NFKB1 in astrocytes which therefore expression more adhesion molecules that increases the accessibility to peripheral lymphocytes is only one potential mechanism. The authors don't present any convincing data that this effect in astrocytes is any more important than the known, and very well established, QTL effect of the variant in immune cells; the authors acknowledge this limitation in the discussion.

Given that the authors have tested a range of outcomes some form of correction for multiple testing across the different assessments as well as within should be considered. The total number of variables considered is quite substantial. For example, the metabolic effects of the associated genotype seem modest and highly unlikely to survive even modest correction.

Finally, while the different approaches employed by the authors have a pleasing and intriguing consistency that supports the authors' contention that MS associated variants might exert effects in CNS cells as well as immune cells, the results need replication. Without a replication effort the marginally significant results in a series of modestly powered experiments are not in themselves totally convincing. The need for replication is particularly strong with regards to the suggestion that the changes in NFKB1 that result from this genotype only influence a subset of NFKB1 target genes. I am sure readers will be happy with the concept that the authors have provided some evidence that this variant might have QTL effects on NFKB1 in astrocytes but the limitations of power and multiplicity of testing make it hard to believe that this might affect just these particular target genes.

Minor issues

- 1) There seem to be some referencing issues. The authors list the same paper as both reference 2 and reference 6. Also this reference only seems to describe 48 MS loci and not 200?
- 2) There are also issues with the figure naming. For example, I could not find Fig. S7 described in the section entitled "Effect of rs7665090G variant on activated human iPSC-derived astrocytes." Similarly, I can't find Fig. S6 etc.

To whom it may concern,

We would like to thank the reviewers for the opportunity to resubmit an improved version of our manuscript.

We have carefully addressed all concerns. In particular, we have now (i) extended our lesion analysis to include microglia/macrophages and endothelial cells, (ii) quantified astrocyte density and damage to glia limitans in MS lesions, and (iii) substantially increased the number of iPSC-derived astrocyte lines, MS lesions and MS patient MRIs used for our analysis. Moreover, we have addressed statistical issues, controlled our analysis for additional NF- κ B relevant risk variants, and clarified questions regarding the function of risk allele-expressing astrocytes in MS pathogenesis. Please find below a detailed point-to-point response to your comments.

We are looking forward to your comments and final decision.

Reviewer #1:

The manuscript „Enhanced astrocyte responses are driven by a genetic risk allele associated with Multiple Sclerosis“ by Ponath et al explores the role of the MS risk allele rs7665090G for astrocytes, CNS resident cells with important functions in MS pathology. Using elaborate in vitro assays combined with post mortem histological analyses, the authors conclusively describe the mechanistic basis for enhanced astrocyte pathology induced by the rs7665090G risk variant, namely enhancement of NF- κ B dependent pathways. The most important findings are that chromatin at the risk locus is accessible in fetal human astrocytes, a prerequisite for rs7665090G driven pathology in astrocytes. Second, the risk variant translates to functional effects mediated by enhanced expression and activity of NF- κ B p50 and p65 as well as their target genes Il15, Icam1, Cxcl10, and Ccl5. Interestingly, in iPSC derived astrocytes, rs7665090G also causes metabolic changes such as decreased glutamate uptake and lactate secretion, pathways relevant for neurotoxicity. Fourth, comparing post mortem MS tissue of risk allele patients to controls, the authors show increased NF- κ B p50/p65 expression and NF- κ B target gene expression in astrocytes as well as enhanced lymphocyte recruitment in chronic active MS lesions. Finally, MRI studies reveal that the risk locus correlates with increased T2 lesion volumes in MS patients.

General critique:

The study presented by Ponath et al presents the relevance of a genetic risk allele for astrocyte biology in a concise and convincing manner; defining the alterations induced by the presence of the risk allele, it underlines the importance of NF- κ B dependent pathways in CNS astrocytes for MS pathogenesis. This work is highly innovative and of high novelty and importance, as it is among the first to define the effects of genetic risk variants in CNS resident cells. Its findings are relevant for both basic scientists studying MS pathogenesis as well as for clinicians treating MS patients.

Thus, I have a few points to consider in a revised version only. If these points are addressed, I recommend publication of this important study.

1. In the results section, ATAC sequencing both from human fetal and iPSC derived astrocytes is mentioned. However, in Figure 1 only fetal astrocytes are shown. Since iPSC derived astrocytes are used for the subsequent studies and the comparability of iPSC derived astrocytes to fetal (or even adult) astrocytes is under scientific debate, showing the comparison of fetal and iPSC derived astrocytes in the ATAC Seq would increase the conclusiveness of the study.

We thank the reviewer for this suggestion. We have added ATAC-seq tracks for iPSC-derived astrocytes to Figure 1. These cells also demonstrate accessible chromatin in the genomic region of interest.

2. How were human fetal astrocytes generated? Inclusion of this information in the Materials and Methods section will be appreciated.

We have obtained the human fetal astrocytes from Thermo Fisher, and have added this information to the material and method section.

3. Please show representative stainings for GFAP and GLAST in the iPSC cultures before and after sorting for GLAST positive cells to underline the effectiveness of the differentiation and selection procedure.

We have now added flow cytometry data for GFAP and GLAST in iPSC-derived astrocytes before and after purification with anti-GLAST coupled magnetic beads (see Supplementary Fig. 12). Our analysis demonstrates that after 40 days of differentiation, 100% of astrocytes expressed GFAP, and 84.8% expressed GLAST before purification. After purification, 100% of astrocytes were GFAP⁺ and 95.3% were GLAST⁺.

4. In Figure 3D, NF- κ B p50 and p65 are globally increased in the risk variant astrocytes both in the cytosol and nucleus. Is there also an increase in the nuclear translocation of NF- κ B in rs7665090G? Analyzing the ratio of nuclear to cytoplasmic NF- κ B p50 and p65 would be interesting to determine potential increased activity of NF- κ B in rs7665090G astrocytes.

The ratio of nuclear to cytoplasmic location in astrocytes does not differ significantly between the risk and protective groups [risk nucleus/cytosol: 0.58 (p50), 0.71 (p65); prot. nucleus/cytosol: 0.77 (p50), 0.97 (p65)]. This indicates that while the genetic risk variant is associated with substantially higher expression levels of p50, it is not associated with an increased nuclear translocation rate. Nevertheless, the absolute numbers of nuclear p50 and p65 are still substantially higher in astrocytes with the risk variant [risk: 45.5/27.2 (p50/p65) vs prot: 16.3/11.3 (p50/p65)].

Minor points:

1. In the results section, Fig S2 is referred to as Fig. S7 in the results section. Please adjust.

We apologize for this oversight, which we have now corrected.

2. Table S5: please use generic names instead of trade names.

Trade names have been replaced with their generic names.

Reviewer #2:

Ponath et al use cellular assays and immunostaining of autopsy material to demonstrate that human derived-astrocytes which express an MS risk variant, rs7665090-G located between the NFKB1 gene and MANBA gene, lead to upregulation of NF-κB expression, protein activity, and target gene expression, as well as markers of A1 reactive astrocytes described by the Barres group. By comparison of iPSC-derived astrocytes and immunostaining from patients homozygous for the risk allele v. protective MS risk variant allele, the authors show that astrocytes plays a critical role in MS pathogenesis through upregulation of a pathway previously implicated in immune cells primarily due to expression of a genetic risk factor allele. However, whether the role of these astrocytes is reactive or causative to lesions seen on MRI and pathology is not directly addressed in this study.

Major issues:

1. Astrocytes are not the only cell that comprises the blood brain barrier; did they also study the MS risk variant effect on pericytes or endothelial cells (at least on autopsy material?) If not, they should still include a short discussion of the potential contributors to the blood brain barrier. Are there any disruptions in astrocyte end-feet morphology in the specimens from patients with the MS Risk variant gene?

This is a very valid point as the risk variant may affect NF-κB signaling in other cell types. We have now extended our analysis of p50/p65 expression to microglia/macrophages and endothelial cells in genotyped MS lesions. The risk variant was associated with increased immunoreactivity of p50 and p65 in CD68⁺ microglia/macrophages with the risk genotype, comparable to astrocytes. In contrast, cytosolic expression of p50/p65 expression did not differ between endothelial cells with the risk and protective genotypes. Thus, the rs7665090 risk variant modulates NF-κB expression in several cell types relevant to lesion formation, including astrocytes, lymphocytes, and macrophages/microglia, but not in endothelial cells. These new findings highlight that the risk variants effects can be cell-type specific. It also substantiates our hypothesis that increased lymphocytic infiltration and lesion sizes in MS lesions with the risk genotype is caused by a combination of enhanced responses by participating cells, which mutually reinforce their effects in individual cell types.

In addition, we have now assessed the astroglial endfeet coverage of blood vessels, and found no difference in disruption of the glia limitans between lesions with the risk and the protective genotypes. We observed a wide spectrum of glia limitans pathology in both groups, suggesting that disruption of astroglial endfeet coverage is not driven by this risk variant, and that the heightened lymphocyte infiltrates in the risk group cannot be explained by astroglial endfeet disruption. We have added this data to the result section (Fig. 3D, E).

2. The one sentence summary says 'astrocyte responses that promote lesion formation' but they do not provide data demonstrating formation of any lesions.

The reviewer rightly points out that we have not directly examined the role of risk variant-driven astrocyte responses in lesion formation, e.g. in a mouse model (EAE). This is because the genetic architecture of non-coding regulatory regions of *NFKB1* differs considerably between humans and mice. The complex genome engineering that is required to replace large genomic fragments in mice by their human counterparts is beyond the scope of this study.

Short of testing the effect in a mouse model, our argument is the following: it is well established that astrocytes are critical contributors to MS lesion formation, by controlling CNS access for peripheral immune cells, by enhancing lesion-promoting functions of other cells, and through direct neurotoxicity [1; 2; 3; 4]. Here we demonstrate that immune factors which mediate the lesion-promoting functions of astrocytes (CXCL10, CCL5, ICAM1, IL15, C3) are upregulated in astrocytes with the risk variant, suggesting that these changes lower the threshold for lesion formation. Our study has therefore linked genetic risk for MS (rs7665090^G) to a specific dysregulation of astrocyte function.

3. Regarding the sentence “the risk haploblock-associated increase in NF- κ B signaling, shown previously in T cells (ref 5), may also apply to astrocytes”. The cited study looks at variants proximal to the NF κ B gene; what is the potential mechanism for how a variant in the region distal to the NFKB1 can alter its expression?

We have investigated the same risk variant (rs7665090^G) that was also examined by Housley et al. in their study [5]. This variant is located distally, not proximally, to *NFKB1*. Cis-regulatory elements are often, but not always, upstream of the transcription site. Moreover, enhancers in intergenic regulatory regions can exert their effect independent of their relative orientation to the activated promoter, i.e. enhancers can be located at long distances upstream or downstream of target genes [reviewed in [6; 7]].

4. Why do patients who are homozygous for the protective allele still develop MS? What is the difference in astrocyte role and function that the authors would predict based on this data?

In complex genetic traits such as MS, not all patients carry all risk alleles, just as unaffected individuals may carry multiple risk alleles. In addition, each individual carries a unique or near-unique combination of risk variants and environmental exposures. This considerable genetic heterogeneity between patients presumably leads to dysregulation of different sets of biological pathways, which in turn might be the cause for the heterogeneity in clinical presentation and treatment responses. For a more detailed discussion of the biological framework of how genetic variation contributes to phenotypic variation, see [8].

5. Page 6, end of paragraph 1 – should cite ‘data not shown’ for non-reactive astrocytes in the lesion vicinity. In addition, it would be helpful to define the density of reactive v non-reactive astrocytes present in the lesions examined, and whether there was a difference between risk variant genotype. How does this compare to MS patients without either allele?

We have now added an analysis of non-reactive astrocytes located in NAWM, which shows that these cells have low baseline expression of p50 and p65, which did not differ between lesions with the risk and protective variants (Supplementary Fig. 6C).

In addition, the density of hypertrophic GFAP⁺ astrocytes in the lesion rim did not differ between the two groups, and was indeed comparable in all examined lesions (Fig. 3F). Similarly, all astrocytes in the active lesion rims were uniformly reactive. The homogenic morphology and density of astrocytes within active rims is consistent with the concept that astrocytes, in contrast to microglia, do not proliferate, migrate, or die in large numbers in MS lesions, but simply tile the underlying white matter [9]. This, however, does not imply functional homogeneity.

Finally, we have not included autopsy cases with the heterozygous genotype in our analysis, because the results from the homozygous cases were highly significant with a clear separation between the two groups.

6. Figure S3 – GFAP staining should be shown separate from as well as merged with the second label (like p65). In Fig 3A, what is the p65⁺ cell identity adjacent to the GFAP⁺p65⁺ cell? How many p65⁺GFAP⁺ v. p65⁺GFAP⁻. p65⁻GFAP⁺ cells (and similarly for all the other markers) are there in the lesions? Rather than densitometric analysis as shown in S3B, the proportion of positive v. negative cells relative to the population of astrocytes in the lesion would be a stronger argument for altered astrocyte signaling in MS risk variant-expressing astrocytes.

We have now changed Supplementary Fig. 6A (formally Figure S3A) to also include the single channel images.

The p65⁺ cell adjacent to the GFAP⁺p65⁺ cell is also a hypertrophic astrocyte albeit with less intense GFAP signal.

Our immunofluorescent data indicates that there were no p50 or p65 negative cells, i.e. that p50 and p65 are expressed constitutively at low baseline levels in all astrocytes, with a wide spectrum of expression in hypertrophic astrocytes. Although we could set an arbitrary threshold for positive immunoreactivity, this would not accurately reflect our results.

7. For the autopsy source patients, which ones were on Tysabri (treatment is not listed in Table S7) – could the protective variant-expressing patients have been on Tysabri, and therefore have fewer lymphocytes in the lesions?

None of the patients from which we derived autopsied CNS tissue were treated with high-efficiency therapies such as Tysabri, Rituxan/Ocrevus or Lemtrada. We have clarified this in Supplementary Fig. 5 (formally Table S7).

8. Figure 4B: are these numbers of T cells normalized (such as to area of lesion?) If they are not normalized, these comparisons may be misleading.

This is a very valid point. We have now normalized the perivascular T cell count to blood vessels and lesion (rim) area. Our data shows that lesions in the risk group contain significantly more infiltrating T cells per blood vessel in the lesion rim, where lymphocyte recruitment is driven by hypertrophic astrocytes, but not in the lesion center, where hypertrophic astrocytes are absent (Fig. 5C, D, formally Figure 4B; and Supplementary Fig. 8). We have further fortified our dataset by analyzing a total of 29 lesions from 10 patients (15 lesions with the risk variant, 14 lesions with the protective variant).

9. The risk variant is only one of many that may be contributing to the phenotype observed. The authors should examine the other NFκB variants mentioned for their composite effect or calculate an NFκB risk score from them.

We agree with the reviewer that other NF-κB relevant risk variants may confound the rs7665090^G-associated astroglial phenotype. Examining the impact of other NF-κB related risk variants on astrocytes to obtain a “NF-κB risk score” would by far exceed the scope of this study. Instead, we have now tested the iPSC lines and MS tissue used in this study for five additional MS risk variants that are predicted to intersect with the canonical NF-κB signaling pathway. The variants and putative gene targets are the surface receptors TNFR1, LTβ-R and CD40 (rs1800693, rs12296430, and rs4810485 respectively), the ligand TNFSF14 (rs1077667) and the intracellular NF-κB adaptor protein TRAF3 (rs12148050). The majority of our iPSC lines and MS tissue was heterozygous for the additional NF-κB relevant risk variants. Homozygosity for additional risk variants was limited to one line or tissue per group. The table for the variant distribution is added as Supplementary Fig. 3. This data suggests that in our experiments, other NF-κB relevant variants do not interfere with the rs7665090 risk variant effect on astrocyte phenotype and lesion pathology.

10. The C3 marker of A1 astrocytes was associated with neurotoxicity and not inflammation. The authors should examine T1 black holes and brain volumes in the same cohort.

T1 hypointense holes and brain atrophy are considered measures of disease severity and possibly propensity to develop progression. As the rs7665090^G variant is associated with susceptibility to MS but not with disease severity, no correlation is predicted between the risk variant and these MRI measures. We have now quantified whole brain volumes, and there was no difference in brain volumes between MS patients with the risk and protective genotype (Supplementary Fig. 10B). Of note, we have added 41 patients to the analysis of both FLAIR and brain volume, and found that the difference in lesion load between MS patients with the risk and protective genotype was no longer significant. We have therefore moved the MRI results to supplementary data (Supplementary Fig. 10A) and discussed these findings vis-à-vis the differences in lesion sizes in our histological analysis.

11. The discussion should be expanded to include how altered astrocytes could ‘promote’ or possibly maintain or worsen lesions in MS. There is no data in this paper that supports astrocytes playing a role in formation of lesions that is worsened by expression of the risk variant. What could be the next steps to understand this role?

We have now expanded our discussion of how variant-associated changes in astrocytes may enhance lesion formation. Please see also our response to comment 2.

12. The markers for A1 astrocytes are increased in risk-variant expressing astrocytes; have the authors addressed whether microglia expressing the MS-risk variant are altered in promoting formation of A1 astrocytes (or whether they are also different within the MS lesion pathology) based on the work of Liddel et al?

This is an excellent point. Our *in vitro* experiments have demonstrated that the risk variant directly impacts on C3 expression in astrocytes via upregulation of NF- κ B signaling. It is possible that enhanced microglia activation in MS lesions with the risk genotype has an additional effect on C3 expression in astrocytes. In fact, this is likely given our new data of increased NF- κ B p50 and p65 expression in lesional microglia. We have now stressed in the discussion that the risk variant acts on multiple cell types within lesions and that this might lead to mutually reinforced responses from these cell types. Co-cultures, e.g. of astrocytes and microglia cells, are a possible way to examine these interactions, but they are beyond the scope of this manuscript.

Reviewer #3:

In this manuscript, Ponath and colleagues investigate the role of the DNA polymorphism rs7665090G on astrocyte function. This variant has been consistently associated with risk to multiple sclerosis, and given the predominantly immune nature of other MS associations, authors wish to explore whether variation at this position alters CNS functions in any way. They start by showing that rs7665090G increases NF- κ B signaling in astrocytes and consequently drives lymphocyte recruitment *in vitro* and in pathological specimens. They also show that this risk allele is associated with increased lymphocyte infiltration within lesions in MS patients. Authors conclude that rs7665090G alters astrocyte function resulting in increased CNS access for peripheral immune cells.

This study is rigorously conducted and conclusions are logically derived from results. Still, some shortcomings were identified:

1. Fig1: The assessment that the chromatin accessibility profiles are “similar” is subjective. Can authors show statistics here (KS test or similar)? They also must show that the profiles for other (non-immune or CNS-related) cell types is substantially different.

We have used ATAC-seq to identify regions of the genome that are accessible, and therefore candidates for further functional studies. Our analysis does not require chromatin accessibility profiles to be “similar” in a quantitative sense, but to demonstrate an open chromatin configuration suggestive of potential gene regulatory function. We have revised the text to clarify that these cells have accessible chromatin in these regions, leaving quantitation of their similarity to subsequent studies performed with larger sample sizes and greater sequencing depth.

We would like to stress that our results that the rs7665090^G variant is associated with differential gene expression in astrocytes bears out the idea that the haplotype block containing rs7665090 is accessible in astrocytes.

Taking these considerations aside, we have calculated the Kolmogorov-Smirnov statistics for the genomic region visualized in Fig. 1 (4:102300000-103200000, GRCh38 coordinates) and found that none of the comparisons reached statistical significance. We examined the cumulative distributions of ATAC-seq reads that are used to calculate the Kolmogorov-Smirnov statistic and found these to be largely concordant across fetal astrocytes, iPSC-derived astrocytes and reference T cell subsets (CD4⁺ T_{reg} and CD4⁺ T_{eff} cells). The largest differences between sample types occurred between T cell subsets and astrocytic samples at 4:102,828,000 and 4:102,871,000. These positions correspond to chromatin accessibility peaks (displayed in Figure 1) that are shared between all samples. This observation suggests that the (non-significant) differences that are apparent between sample types are driven by quantitative differences in read depth at ATAC-seq peaks, rather than different overall patterns of accessibility in this region. While these results support our original contention of “similarity” across cell types, the Kolmogorov-Smirnov statistic may be overly sensitive to small, sub-nucleosomal shifts in the position of ATAC-seq peaks and/or differences in read coverage across samples. As our dataset

is not powered to address these issues, we have revised the manuscript to emphasize the presence of chromatin accessibility in all samples, rather than the quantitative similarity between these profiles.

2. Fig 2: no correction of multiple comparisons (n=84) is attempted. It looks like only IL15 would survive this. Thus, if this is the only evidence presented, conclusions on the involvement of C3, while potentially interesting, remains speculative.

This is a very valid comment. After correction for multiple comparisons using the Benjamini-Hochberg procedure, none of the NF- κ B gene targets remained significant. We have therefore increased the number of patient-derived astrocyte lines to 6 lines per group, i.e. a total of 12 lines. With this improved statistical power, differential expression was statistically significant in the same set of NF- κ B gene targets after FDR correction.

3. Fig 2: Fig 2E repeats information from fig 2A. Another volcano plot at protein level would represent the same information and reduce the Figure size by one panel.

In order to reduce the figure size, we have omitted Fig. 2E, as this information is already represented in Fig. 2D. Since Fig. 2F (now Fig. 2E) contains only 7 proteins, the information might be better captured with a bar graph rather than a volcano plot.

4. If 18% of MS associations affect NF- κ B, isn't it surprising to see such a large effect with just 1 variant?

This is an excellent point. Our results are in line with other studies that also reported sizable variant effects, e.g. rs228614^G is associated with a ~20 fold upregulation of NF- κ B p50 expression and a ~2.5 fold increase in nuclear translocation of pp65 in PBMCs and CD4 cells. Similarly, rs1800693^G is associated with ~5 fold increase in nuclear translocation of pp65 [5] and with several-fold upregulation of genes such as CD40 and CYP27B1 [10].

We would like to slightly qualify our statement that 18% of MS risk variants are predicted to intersect with the NF- κ B pathway. This includes both upstream regulators and downstream effectors, and not all variants may necessarily impact on NF- κ B expression or signaling itself. As pointed out in response to comment 9, reviewer 2, we have now controlled for five relevant NF- κ B relevant variants in iPSC-derived astrocytes and in MS tissue (Supplementary Fig. 3).

5. Where else (outside of astrocytes) might this variant have an effect? Their observation that enhancer immunoreactivity for NF- κ B in lesions with the risk genotype was not restricted to astrocytes (also found in GFP- cells) supports the possibility that this variant affects NF- κ B function in a much broader set of cells.

We agree with the reviewer and have extended our analysis of NF- κ B subunit p50 and p65 immunoreactivity to two more cell types, microglia/macrophages and endothelial cells, in genotyped brain tissue. The results indicate that the risk variant is associated with increased NF- κ B expression in microglia/macrophages but not in endothelial cells (Fig. 4E-H). This highlights that genetic variants may effect several but not all cell types. We have stressed in the discussion that the risk variant acts on astrocytes, microglia, and lymphocytes within lesions and that this might lead to enhanced and mutually reinforced responses from these cells types.

Reviewer #4:

In their expression analysis the authors find that the level of p50 is higher in iPSC derived astrocytes from risk allele homozygotes (n=3) than in protective allele homozygotes (n=3), and also find modestly significant differential expression for a few of the 84 NFKB1 target genes they tested.

1. While these are interesting and novel data, I am not sure that the authors suggestion that "the rs7665090 risk variant activates a SPECIFIC SET of NFKB target gene" can be justified.

We would like to modify our statement and suggest that the rs7665090 risk variant results in differential expression of a particular set of NF- κ B target genes under specific stimulatory conditions (response eQTL). Given that the addition of data from 6 additional iPSC-derived astrocyte lines did not broaden the number of differentially expressed NF- κ B target genes, this seems a valid statement. We have added this point to the discussion.

2. Given that the study subjects will inevitably have varied at MS risk genotypes other than rs7665090 it could be that effects on other NFKB1 target genes have been confounded.

We agree with the reviewer that this is a very valid concern. We have now genotyped iPSC-derived astrocytes and MS autopsy cases for MS risk variants predicted to directly impact on the canonical NF- κ B signaling pathway. The risk variants and putative gene targets are the surface receptors TNFR1, LT β -R and CD40 (rs1800693, rs12296430, and rs4810485 respectively), the ligand TNFSF14 (rs1077667) and the intracellular NF- κ B adaptor protein TRAF3 (rs12148050). We found that the additional NF- κ B relevant risk variants were approximately equally distributed in the rs7665090 risk and protective groups, with the majority of iPSC lines and MS tissue being heterozygous for the risk variants (Supplementary Fig. 3). This data suggests that in our experiments, other NF- κ B relevant variants do not interfere with the rs7665090^G effect on astrocyte phenotype and lesion pathology (see also response to reviewer 2, comment 9).

3. Similarly, the modest power of their analysis, which included just 6 subjects, is also likely to have confounded the chances of seeing effect on other NFKB1 target genes. Previous expression studies have highlighted a prominent eQTL effect of this variant on MANBA expression but have not highlighted effects on NFKB1. This could of course be an astrocyte specific effect on NFKB1 but it's a shame the authors did not measure the expression of MANBA in these astrocytes.

We have now added iPSC-derived astrocyte lines from 6 additional individuals to improve the statistical power for detecting changes in NF- κ B target gene expression after stimulation. With this increased statistical power, the number of differentially regulated NF- κ B target genes have not increased. However, we cannot exclude that a larger sample size will yield additional genes with pronounced expression shifts. Moreover, we think that it is highly likely that under different stimulatory conditions additional/other target genes are differentially regulated by the rs7665090 risk variant (see also response to comment 1 by reviewer 4).

We have also quantified mRNA expression of MANBA in iPSC-derived, IL-1 β /TNF α stimulated astrocytes, and found comparable levels of expression in both groups (Supplementary Fig. 4D). This suggests that the rs7665090^G variant impact on MANBA expression is cell type-specific and/or stimulus-dependent.

4. In their pathological assessments the authors observed equivalent changes in hypertrophic astrocytes from carefully selected lesions examined histologically in 10 biopsied cases. However, they do not seem to have tested other NFKB1 target genes from their list of 84; at least they only present data for those genes highlighted in their iPSC expression analysis.

Here we respectfully disagree with the reviewer, as we have quantified the NF- κ B target genes CXCL1 and CCL2 in lesional astrocytes, which were not differentially regulated in iPSC expression analysis (Fig. 4D and Supplementary Fig. 6A, B). We have highlighted these data in the result section. In addition, we have quantified expression of two astroglial activation markers, GFAP and iNOS, that are not regulated by NF- κ B and that were also not differentially regulated in the two groups in iPSC-derived and lesional astrocytes.

5. The authors also report nominally significant associations of the rs7665090 genotype with pathological and MRI features. However, these are only nominally significant and I am not exactly sure I follow the logic behind excluding patients with low lesion load and short duration of disease. I can't see why these should be biased with respect to genotype. The authors should include an analysis based on the scans from all 93 subjects at least to reassure the reader that the excluded group did not include an over representation of the risk allele carriers just by chance. Even if the

genotype associations with pathology and MRI features were confirmed this would not imply that these effects were driven by NFKB1 changes in astrocytes.

We acknowledge that the exclusion criteria were somewhat artificial. We have subsequently added subsequently 41 more patients to our T2 FLAIR MRI study for a total of 134 patients (78/56 risk/protective) to improve the statistical power of our findings. This has resulted in a loss of significance for the difference in lesion load between MS patients with the risk and protective genotype. We have moved these negative results to the supplementary data (see Supplementary Fig. 10A) and discussed these findings vis-à-vis the differences in lesion sizes in our histological analysis.

6. The mechanism suggested by the author, that the risk allele increases the expression of NFKB1 in astrocytes which therefore expression more adhesion molecules that increases the accessibility to peripheral lymphocytes is only one potential mechanism. The authors don't present any convincing data that this effect in astrocytes is any more important than the known, and very well established, QTL effect of the variant in immune cells; the authors acknowledge this limitation in the discussion.

As the reviewer points out, we do not suggest that the rs7665090^G variant confers MS risk exclusively through changes in astrocyte function, or that the astrocyte-mediated MS risk is more important. A key message of our study is that we have linked genetic risk for MS (rs7665090^G) with astrocyte dysfunction. Thus, MS may result not only from variant-driven dysregulation of the peripheral immune system but also from perturbed responses in the CNS.

7. Given that the authors have tested a range of outcomes some form of correction for multiple testing across the different assessments as well as within should be considered. The total number of variables considered is quite substantial. For example, the metabolic effects of the associated genotype seem modest and highly unlikely to *survive even modest correction*.

This point is well taken. We have now increased the number of iPSC-derived astrocyte lines and performed FDR correction using the Benjamini-Hochberg procedure. With this, the increase in gene expression remained significant (see reviewer 3, point 2).

We have now clarified that the reported p-values for the glutamate uptake and lactate release experiments are not from a student's t-test but from a Tukey-Kramer multiple comparison *test performed after one-way ANOVA, a method that accounts for multiple samples and does not require further corrections*. In the figures, we have only reported p-values for differences between the risk and protective groups, which may erroneously suggest that we have used a student's t-test.

8. Finally, while the different approaches employed by the authors have a pleasing and intriguing consistency that supports the authors contention that MS associated variants might exert effects in CNS cells as well as immune cells, the results need replication. Without a replication effort the marginally significant results in a series of modestly powered experiments are not in themselves totally convincing. The need for replication is particularly strong with regards to the suggestion that the changes in NFKB1 that result from this genotype only influence a subset of NFKB1 target genes. I am sure readers will be happy with the concept that the authors have provided some evidence that this variant might have QTL effects on NFKB1 in astrocytes but the limitations of power and multiplicity of testing make it hard to believe that this might affect just these particular target genes.

We have now doubled the number of iPSC-derived astrocyte lines to 6 lines per group, and have increased the number of MS lesions used for quantification of lymphocytic infiltrates from 10 to 29. This added statistical power has not increased the number of NF-κB gene targets that were differentially expressed under specific conditions in astrocytes. We cannot exclude that a larger sample size will yield additional genes with pronounced expression shifts (see also response to comment 3).

Minor issues

1) There seem to be some referencing issues. The authors list the same paper as both reference 2 and reference 6. Also this reference only seems to describe 48 MS loci and not 200?

We apologize for this oversight, which we have now corrected.

2) There are also issues with the figure naming. For example, I could not find Fig. S7 described in the section entitled "Effect of rs7665090G variant on activated human iPSC-derived astrocytes." Similarly, I can't find Fig. S6 etc.

We have now included the missing supplementary data.

References

- [1] G. Ponath, C. Park, and D. Pitt, The Role of Astrocytes in Multiple Sclerosis. *Frontiers in Immunology* 9 (2018) 217.
- [2] M.V. Sofroniew, Astrocyte barriers to neurotoxic inflammation. *Nature reviews. Neuroscience* 16 (2015) 249-263.
- [3] M.A. Wheeler, and F.J. Quintana, Regulation of Astrocyte Functions in Multiple Sclerosis. *Cold Spring Harbor perspectives in medicine* (2018).
- [4] S.A. Liddelow, K.A. Guttenplan, L.E. Clarke, F.C. Bennett, C.J. Bohlen, L. Schirmer, M.L. Bennett, A.E. Münch, W.-S. Chung, T.C. Peterson, D.K. Wilton, A. Frouin, B.A. Napier, N. Panicker, M. Kumar, M.S. Buckwalter, D.H. Rowitch, V.L. Dawson, T.M. Dawson, B. Stevens, and B.A. Barres, Neurotoxic reactive astrocytes are induced by activated microglia. *Nature* 541 (2017) 481-487.
- [5] W.J. Housley, S.D. Fernandez, K. Vera, S.R. Murikinati, J. Grutzendler, N. Cuerdon, L. Glick, P.L. De Jager, M. Mitrovic, C. Cotsapas, and D.A. Hafler, Genetic variants associated with autoimmunity drive NFκB signaling and responses to inflammatory stimuli. *Science translational medicine* 7 (2015) 291ra93-291ra93.
- [6] M. Bulger, and M. Groudine, Functional and mechanistic diversity of distal transcription enhancers. *Cell* 144 (2011) 327-39.
- [7] J.L. Plank, and A. Dean, Enhancer function: mechanistic and genome-wide insights come together. *Molecular cell* 55 (2014) 5-14.
- [8] N.R. Wray, C. Wijmenga, P.F. Sullivan, J. Yang, and P.M. Visscher, Common Disease Is More Complex Than Implied by the Core Gene Omnigenic Model. *Cell* 173 (2018) 1573-1580.
- [9] M.V. Sofroniew, and H.V. Vinters, Astrocytes: biology and pathology. *Acta neuropathologica* 119 (2010) 7-35.
- [10] L. Ottoboni, I.Y. Frohlich, M. Lee, B.C. Healy, B.T. Keenan, Z. Xia, T. Chitnis, C.R. Guttman, S.J. Khoury, H.L. Weiner, D.A. Hafler, and P.L. De Jager, Clinical relevance and functional consequences of the TNFRSF1A multiple sclerosis locus. *Neurology* 81 (2013) 1891-9.

Reviewers' Comments:

Reviewer #1:

Remarks to the Author:

Ponath et al. have revised their manuscript "Enhanced Astrocyte Responses are Driven by a Genetic Risk Allele Associated with Multiple Sclerosis" and have addressed my concerns and issues to my full satisfaction.

I have no further questions and thus recommend publication.

Reviewer #2:

Remarks to the Author:

The authors have responded with fairly extensive revisions that address most of the concerns that were raised. Of note, the MRI lesion load association failed to confirm in a larger sample set so this was relegated to the supplement. Importantly, the risk variant was also associated with p50 and p65 immunoreactivity in CD68+ microglia/macrophages but not in endothelial cells.

I suggest the one sentence summary be modified, since the authors did not show that this MS risk gene "promotes lesion formation" rather just was modestly associated with lymphocytic infiltrates (N=5 per group), to one of the other conclusions they make throughout the paper i.e. "...genetic risk variant is linked to dysregulation of astrocyte function" or "is associated with increased expression of genes in astrocytes that are known to be important in lesion formation/amplification".

Similarly, in the abstract, the line stating that the risk variant linked with ".....dysfunctional astrocyte responses, which result in increased CNS access for peripheral immune cells" would be best modified to "dysfunctional astrocyte responses associated with" since pathological studies cannot prove causation, just show an association

Reviewer #3:

Remarks to the Author:

Authors have made a significant effort to address all my questions. I am satisfied with the approach followed to address the question of whether the chromatin accessibility profiles are sufficiently similar or not. Ultimately, I agree that a qualitative assessment is sufficient to prove their point, and agree with the modifications to the text.

I was impressed by their initiative to analyze more patient-derived astrocyte lines, an approach that ultimately strengthened their results.

Reviewer #4:

Remarks to the Author:

The authors have appropriately addressed the concerns raised and have actively engaged in the review process. I have no further issues

1. Please address the remaining comments of Reviewers with textual changes (below).

Reviewer 1,3 and 4 had no further questions.

REVIEWER 2:

I suggest the one sentence summary be modified, since the authors did not show that this MS risk gene “promotes lesion formation” rather just was modestly associated with lymphocytic infiltrates (N=5 per group), to one of the other conclusions they make throughout the paper i.e. ‘....genetic risk variant is linked to dysregulation of astrocyte function” or “is associated with increased expression of genes in astrocytes that are known to be important in lesion formation/amplification”. Similarly, in the abstract, the line stating that the risk variant linked with “.....dysfunctional astrocyte responses, which result in increased CNS access for peripheral immune cells” would be best modified to “dysfunctional astrocyte responses associated with” since pathological studies cannot prove causation, just show an association

We have modified both the abstract and the one sentence summary according to reviewer's suggestions.